# Simulation metamodeling approach to complex design of garment assembly lines

Ocident Bongomin[1]*, Josphat Igadwa Mwasiagi[1], Eric Oyondi Nganyi[1], Ildephonse Nibikora[2]

1 Department of Manufacturing, Industrial and Textile Engineering, School of Engineering, Moi University, Eldoret, Kenya, 2 Department of Polymer, Industrial and Textile Engineering, Faculty of Engineering, Busitema University, Tororo, Uganda

* ocidentbongomin@gmail.com

**Data Availability Statement:** Data are available from Mendeley (dx.doi.org/10.17632/t5w96kh5w7).

**Funding:** The author OB received funding (Credit No. 5798-KE) from Africa Center of Excellence II in Phytochemicals, Textiles and Renewable Energy

## Abstract

The today's competitive advantage of ready-made garment industry depends on the ability to improve the efficiency and effectiveness of resource utilization. Ready-made garment industry has long historically adopted fewer technological and process advancement as compared to automotive, electronics and semiconductor industries. Simulation modeling of garment assembly line has attracted a number of researchers as one way for insightful analysis of the system behaviour and improving its performance. However, most of simulation studies have considered ill-defined experimental design which cannot fully explore the assembly line design alternatives and does not uncover the interaction effects of the input variables. Simulation metamodeling is an approach to assembly line design which has recently been of interest to researchers. However, its application in garment assembly line design has never been well explored. In this paper, simulation metamodeling of trouser assembly line with 72 operations was demonstrated. The linear regression metamodel technique with resolution-V design was used. The effects of five factors: bundle size, job release policy, task assignment pattern, machine number and helper number on throughput of the trouser assembly line were studied. An increase of the production throughput by 28.63% was achieved for the best factors' setting of the metamodel.

## Introduction

The disruptive transformation in industrial sectors is being experienced in garment and textile manufacturing more rapidly than most sectors [1]. This is because garment and textile industries are among the oldest sectors that have received very little technological advancement. For this reason, they are experiencing disruptive technological leapfrogging and enormous competition in the business environment in the era of industry 4.0 [2]. The disruptive transformation in textile and garment industry is well-known today as Fashion 4.0 or Apparel 4.0 [1]. In fact, most countries are currently revitalizing and retrofitting all their manufacturing sectors including garment industry in order to harness the sustainable competitiveness [3]. Therefore, for the garment industry to remain competitive, it must be able to satisfy customers' demand

(ACE II-PTRE) of Moi University, https://
excellencecenter.mu.ac.ke/. The funders had no
role in study design, data collection and analysis,
decision to publish, or preparation of the
manuscript.

**Competing interests:** The authors have declared
that no competing interests exist.

by improving the line efficiency and productivity by adopting advanced assembly line design techniques such as simulation, metamodeling and optimization [4].

Simulation is one of the key disruptive technologies that has remained emblematic of industry 4.0 although it is an old technology [5, 6]. It has been majorly applied in the analysis of complex systems to give extensive insights into systems' behaviour. In assembly line design, simulation modeling has been used to generate design alternatives and to enable tactical and strategic management of the stochastic nature of assembly or production systems [7]. Simulation models are basically classified as continuous, discrete event, discrete/continuous (hybrid), and Morte Carlo simulations [8]. The choice of simulation models is often based on functional characteristics of the system and the objectives of the study. Discrete event simulation represents only the points in time at which the state of the system changes. This means that the system is modelled as a series of events, that is, instants in time when a state change occurs [9, 10]. Discrete event simulation has been used extensively in garment manufacturing and other manual or semi-automatic production systems such as footwear, electronics and automotive assembly lines [11]. For instance, Guner & Unal [12] investigated the application of computer simulation for the design of a manufacturing process for T-shirt production in a virtual-reality environment. Recent studies have demonstrated the feasibility and suitability of using discrete event simulation technique for assembly line design in garment industry [13, 14].

Unfortunately, simulation modeling of a garment assembly line is a very complex task because it comprises of many hard-to-predict variables which have to be considered. In addition, software for complex assembly line design is computationally intensive (for example Arena, Simul8, Anylogic, and Enterprise dynamics) [15], and most scenarios-based simulations are typically performed with complex "black box" models with many variable design parameters in which the users normally have no clear understanding of the underlying equations and how the inputs interact with each other [16, 17]. This makes computer simulation very tedious and impractical to run thousands of simulations for thorough design space exploration, sensitivity analysis and optimization. This computational limitation of simulation modeling can be overcome by incorporating metamodels [16, 18, 19].

Metamodels, commonly known as surrogate models, response surfaces, approximate models or emulators are used to approximate the input-output behavior of simulation models [20]. The term indicates a mathematical approximation that models the behavior of another model [18, 21]. They have been used in several fields of research to contravene runtime issues with analyzing and experimentation of computational demanding simulation models [22]. For instance, regression techniques of metamodeling have been used to estimate relationships between model inputs and outputs based on results of a probabilistic sensitivity analysis [21–23]. More specifically, metamodels build a closed-form mathematical expression to approximate the input and output relationship implied by the simulation model based on simulation experimental runs at selected design points in advance [24, 25]. This can be easily evaluated in a spreadsheet environment "on demand" to answer what-if questions without the need to run lengthy simulations [26]. In optimization, the main advantages of metamodel include improving the efficiency of optimization, supporting parallel computation, covering sensitivity analysis of input variables and gaining better insights into the problem, and handling both discrete and continuous variables [19].

The most commonly used approaches for metamodel construction are statistic-based and machine-learning. The former solely depends on the data received from the simulation experiments which includes linear (polynomial) regression, support vector regression, multivariate adaptive regression spline, Gaussian process regression (kriging), and radial basis function [19, 27–31]. The latter is based on neural networking, rule learning, and fuzzy logic [16, 18, 32, 33]. For example, Haefner et al. [34] applied machine learning approach based on artificial

neural network for developing metamodel to be used for tooth root stress analysis of micro-gear. Morin et al. [35] used machine learning to generate metamodels for sawing simulation in wood industry. Altogether, metamodels transform the implicitly stochastic response of the simulation as an explicit deterministic functional form [24, 27, 29]. Kleijnen & Sargent [36] suggested ten (10) steps for developing the linear regression (including polynomial) metamodels for random simulation. The three main steps include: choosing a functional form for the metamodeling function based on the study goal, designing and executing the experiments to fit the metamodel, and model learning/fitting the metamodel and validating the quality of its fit [26, 37]. Designing and execution of simulation experiments is one of the fundamental steps in regression metamodeling [38]. A fractional factorial design (e.g. Resolution-V design) is very suitable for designing simulation experiments [39]. It is basically generated from a full factorial experiment by choosing an alias structure. It overcomes the limitation of the full factorial design by screening some factors, and focusing on the main factors [40]. Moreover, it has the ability to separate main effects and low-order interactions [38]. In addition to the classical design of experiments including factorial and central composite designs, the space filling design approaches (Latin hypercube sampling and orthogonal array) have also been used for simulation experiment designs [19, 41].

Metamodels have got many real-world application areas. For example, metamodeling has already been applied in health economics [22]. It has also found a potential application in agriculture such as statistical modeling of maize [42]. Further, it has been employed in the analysis of building structures [31], in cyber-physical systems [43] and grid system low carbon energy technology [30]. Despite the numerous applications of metamodeling, its application for designing assembly line has not been explored. However, two types of ill-designed experiments have been used by the previous studies. The first type occurs when the analysts perform scenario-oriented experiments, where putting the focus on pre-selected interesting combinations or a trial-and-error approach which is time consuming but does not address the fundamental questions [44]. The second one occurs when the researchers or analysts start with a baseline-scenario and vary one factor at a time while in practice, the factors are likely to interact. Therefore, if there are factor interactions, varying one factor at a time will never uncover them [45]. Previous studies indicated that most simulation experiments have been ill-defined for garment assembly line design [7, 14]. Therefore, simulation metamodeling was proposed in the present study as a technique to attain an inexpensive evaluation of garment assembly line design and investigate the effect of factors on throughput which contribute to improvement of decision making at operational and tactical production planning. This paper demonstrates the applicability and suitability of designing a complex garment assembly line using simulation metamodeling.

## Methodology

The empirical data to support the simulation modeling was collected from Southern Range Nyanza Limited (NYTIL) garment manufacturing facility (Jinja, Uganda) with the objective of improving the throughput of trouser assembly line. Industrial engineering tools: time study, process mapping, fishbone diagram, brainstorming and observations were used to obtain the empirical data. The study was conducted in four phases including identification of input variables, conceptual modeling, simulation model development and metamodeling as schematized in Fig 1. The methodology phases of the study are described in the penultimate subsections.

### Identification of input variables

The input variables that influence the throughput of the garment sewing line were identified by brainstorming four categories of people in garment production department namely;

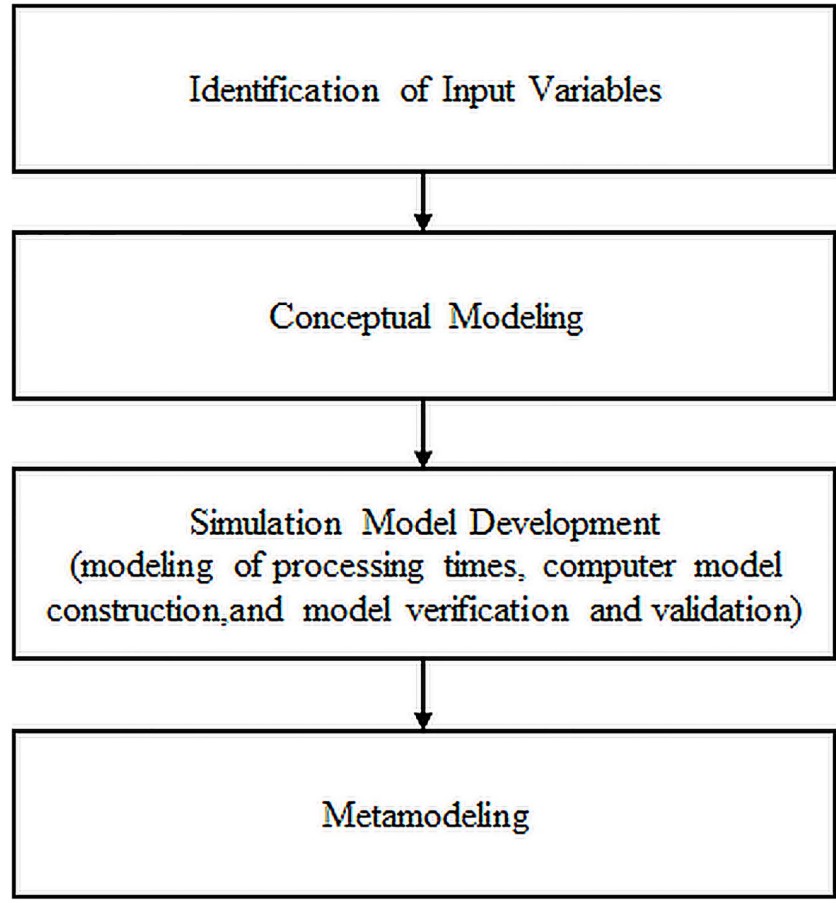

**Fig 1. Methodology approach for this study.**

operators, quality personnel, maintenance personnel and line supervisors. The brainstorming was conducted on an individual basis during their free time to avoid interrupting production. All their ideas were collected and categorized using fishbone diagram based on the Big four major category of causes (4M) in a manufacturing system: Manpower, Method, Material and Machine as depicted in Fig 2.

## Conceptual modeling

To this end, all processes involved in trouser assembly line were summarized using the conceptual model. It is a series of logical relationships relative to the components and structure of trouser assembly line. The conceptual modeling involved mapping all the processes or tasks associated with making trousers. In order to capture all trouser assembly line processes, the assembly line was broken down into 10 preparation sections and main body assembly section. The preparation sections (subassembly processes) included adjustable, knee flap, knee pocket, hip flap, back, front, big loop, small loop, back patch, and side pocket and flybox preparation (Fig 3). Where 1–72 represent the tasks performed by the machine operators or helpers and a-q are the trouser parts to be assembled. Validation of the conceptual model was done through comparison between process mapping and the real-world trouser assembly line. The

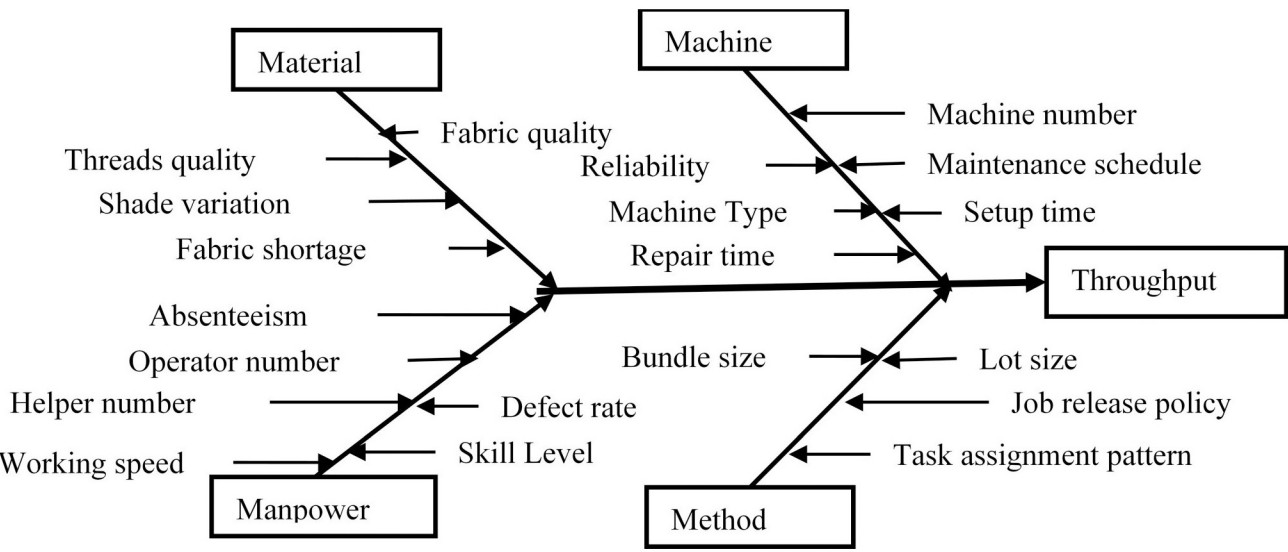

**Fig 2. The fishbone (cause-and-effect) diagram.**

conceptual model of the trouser assembly line was verified by the line supervisors and workers in another department.

## Simulation model development

**Modeling of processing times.** The processing times data is the heart of Arena simulation modeling and was obtained using continuous stopwatch time study combined with observations method [46]. The time study was conducted at three intervals of different production seasons each with 20 measurements per task. Therefore, a total of 60 measurements per task were obtained for analyses so as to capture most variabilities in the processing times. The observed task times for a part (cut piece) in a bundle was multiplied by the total number of cut pieces in a bundle to estimate the bundle processing times. For example, the processing times for bundle size 25 (number of cut pieces in a bundle). Arena input analyzer was used to analyze these processing times so as to obtain the candidate probability distributions and the fitted probability distribution. The examples of the fitted processing time probability distribution for some of the tasks: knee patch attach and buttonhole on left flybox are given in Fig 4. The fitting of the processing times was done for all tasks involved in the trouser assembly line. The fitted processing time probability distribution for each task was then used in building discrete event simulation model. The processing time distribution of different garment bundle sizes 10, 25 and 40 are presented in S1, S2 and S3 Tables, respectively.

**Computer model construction.** The computer model of the trouser assembly line was constructed using Arena simulation environment. Two categories (32 and 64 bits) of the Arena simulation software (academic license version 16, Rockwell Automation Inc., USA) were obtained. The 32 bits Arena software was chosen and installed on a low processing speed Lenovo V110 notebook computer with 64 bits, 2.00 GHz Intel Core i3 CPU and 4.00 GB RAM. The computer modeling of preparatory sections of trouser assembly line including front and back were done separately and then combined using Arena Match modules to form one complex trouser assembly line model. The four trouser assembly line models included main body assembly (MBA), front (FP), back (BP) and big loop (BLP) as shown in Fig 5. The project bar is the most important user interface of Arena simulation environment for building

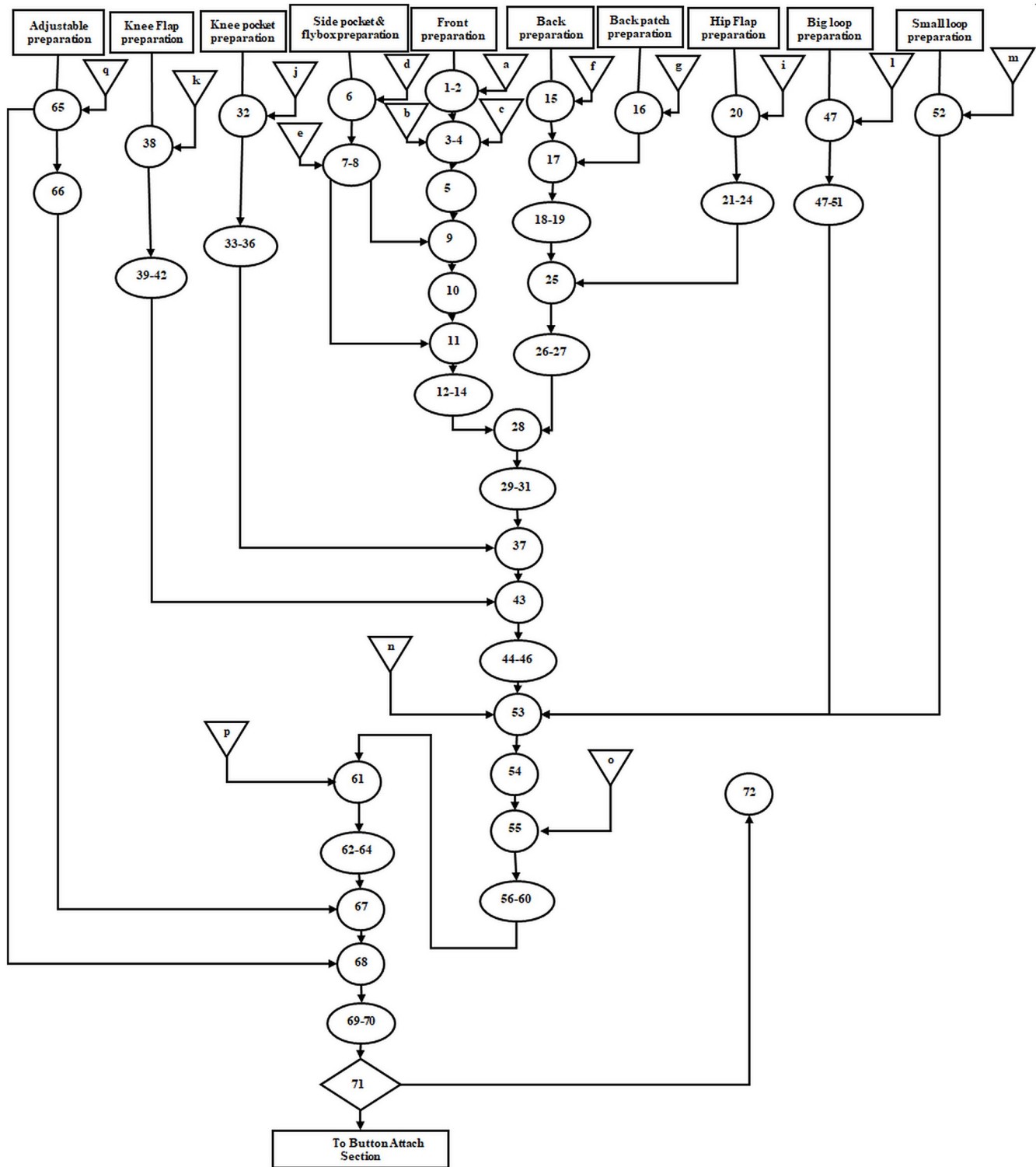

**Fig 3. Conceptual model of trouser assembly line.** a- left flybox, b- leg front, c- knee patch, d- right flybox, f- leg back, g- back patch, h- hip pocket, i- hip flap, j- knee pocket, k- knee flap, l- big loop, m- small loop, n- waist band, o- company tags and size label, p- bottom leg rope, q- adjustable rope.

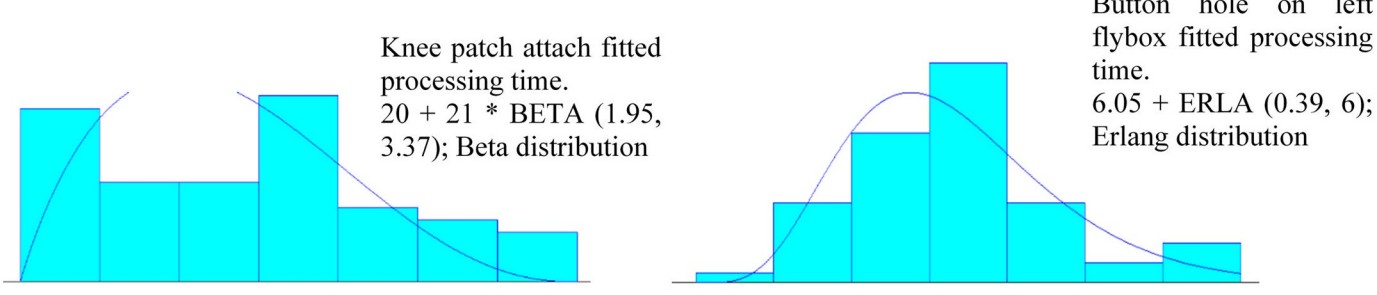

Knee patch attach fitted processing time.
20 + 21 * BETA (1.95, 3.37); Beta distribution

Button hole on left flybox fitted processing time.
6.05 + ERLA (0.39, 6); Erlang distribution

**Fig 4. Fitted processing time probability distributions from Arena input analyzer.**

simulation model. Therefore, three elements from the Arena project bar including basic process, advanced process and transfer were used. For further understanding of the simulation model of the trouser assembly line, the constructed Arena simulation model (base model) is deposited at https://doi.org/10.17632/T5W96KH5W7.1 [47].

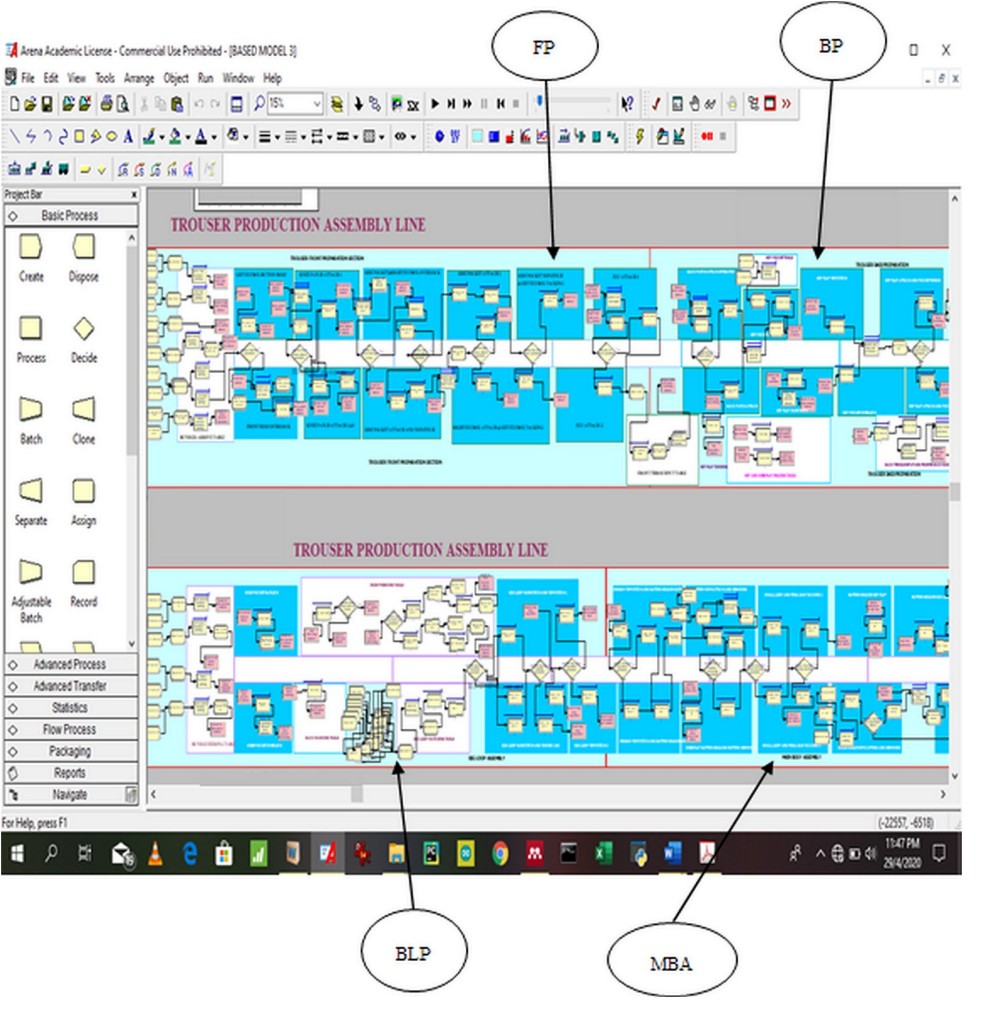

**Fig 5. Arena simulation model of trouser assembly line.**

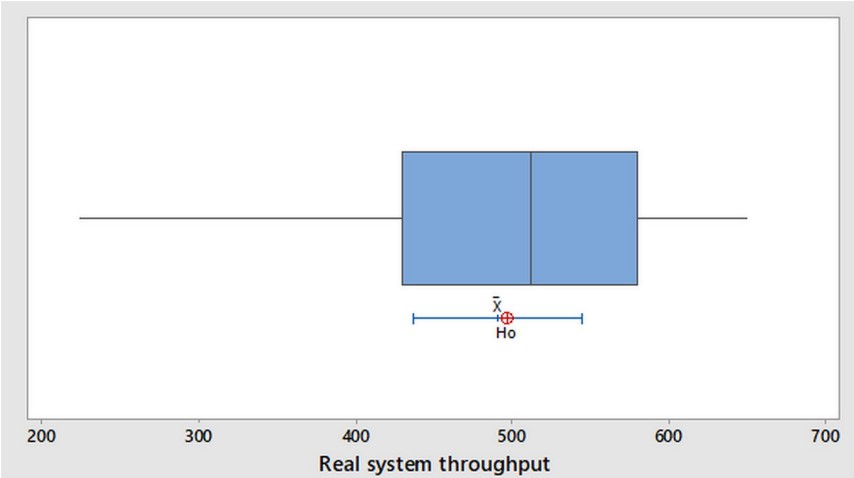

**Fig 6. Boxplot of real system throughput with $H_0$ and 95%t-CI.**

**Model verification and validation.** The simulation model was verified using traces and animation technique (S1 Fig). The simulation replication number (n = 10) was determined as used by Kelton et al. [48]. Steady-state simulation with 2 days warm-up period was approximated according to Law [49]. Due to the complexity of the trouser assembly line simulation model, a run length of one month (28 days of 8 hours daily production) was used. The simulation runs were executed, and the line production throughput ($\mu_A$ = 496 pieces per day) with half width (6.61) was achieved (S2 Fig). The hypothesized mean ($\mu_A$) was used for comparison with real-world system (trouser assembly line) throughput samples. The line production throughput with sample size (N = 23) collected for a period of one month was used to validate the operation of the trouser assembly line simulation model (S4 Table). One-sample-T hypothesis test at 95% confidence interval (CI) was done using Minitab Statistical Software (version 18, Minitab Inc., USA). The null hypothesis ($\mu_0$) was accepted because $\mu_A$ lies within 95% CI for real-world system average throughput ($\mu_R$ = 490 pieces per day) with the T-value (-0.2) and P-value (0.842) as depicted in Fig 6.

## Metamodeling

The metamodeling process was conducted according to Wallach [42], following the four steps approach which included definition of experimental design, generation of training datasets, model training/fitting and validation as schematized in Fig 7.

**Definition of experimental design.** The definition of the simulation experimental design is the first step in metamodeling. The dependent variable (throughput) was determined to meet the objective of the study. The independent variables (input factors) are very critical. Thus, five factors were selected from the fishbone diagram. These factors were those defined by the team that brainstormed. It was agreed by the team that these factors were the most significant ones that contributes to production throughput of the garment assembly line. All the factors were studied at two levels (i.e. low and high) as described below.

Factor A (Bundle size). This is the number of cut pieces of each part of the trouser (or any other woven garment product) which are moved from one operator to another [50]. Different bundle sizes were being used in garment manufacturing. Therefore, two levels:10 and 40 were used in this study to determine their effect on the overall throughput of the production line.

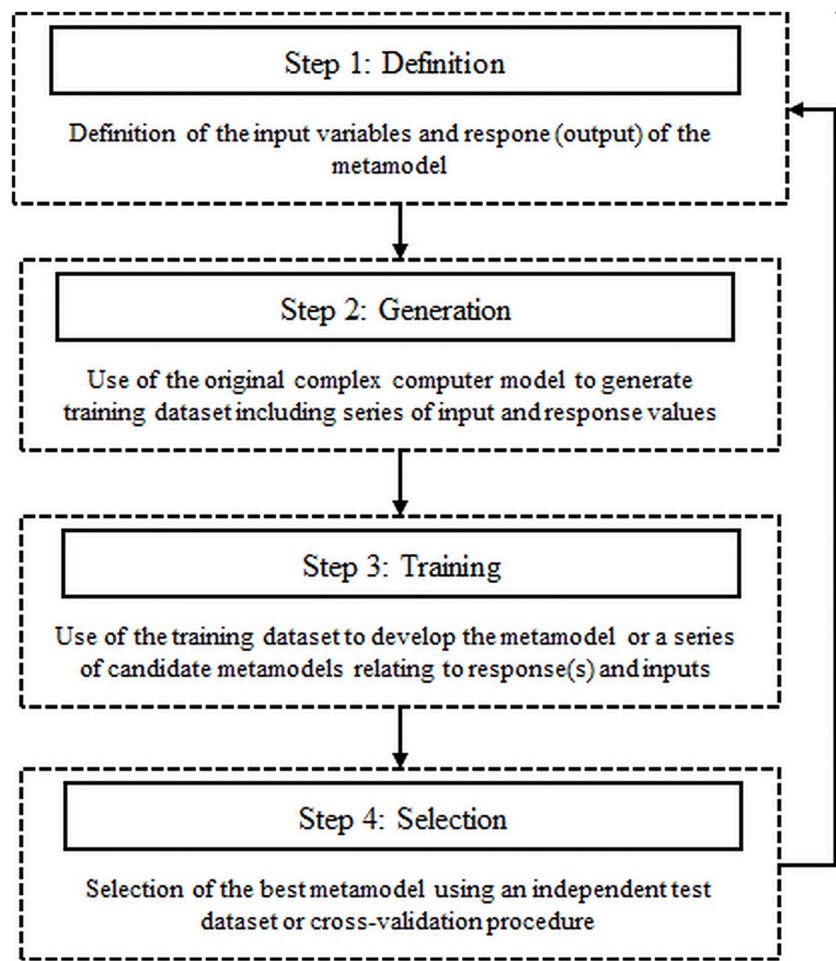

**Fig 7. Regression metamodeling approach.**

Factor B (Job release policy). This is the method of availing input materials into the production line. The effect of its two levels (no policy and policy) on the throughput were studied. No policy level meant the input materials are made available to the production line at a constant rate i.e. everyday. While for policy level, the input materials are made available depending on the work in progress (WIP) threshold of the bottleneck workstation.

Factor C (Task assignment pattern). This is the method of distributing workload to the operators performing the same tasks in the workstations. The two levels studied were random and equal task assignment pattern. With the random task assignment, the workload of operators performing similar tasks are randomly distributed while in equal task assignment pattern, the workload of operators performing similar tasks are equally distributed.

Factor D (Machine number). In the present case, the machine number was considered as a categorial factor because of the different machine types and several workstations involved in the trouser assembly line. Therefore, the machine number to be varied was determined for the bottleneck and idle workstations. Also, it was studied at two levels: increase and decrease. In the case of increase level, three single needle lockstitch and one iron press machines were added in the production line. In the decrease level, three single needle lockstitch and one buttonhole machines were removed from the production line.

**Table 1. Experimental design specification.**

| Factors | Level | Base design | Resolution | Run | Replicates | Fraction | Blocks | Center point |
|---------|-------|-------------|------------|-----|------------|----------|--------|--------------|
| 5 | (- /+) | 5,16 | V | 16 | 1 | ½ | 1 | 1 |

Design generator; E = ABCD, Defining relation; I = ABCDE, Alias structure; I + ABCDE, A + BCDE, B + ACDE, C + ABDE, D + ABCE, E + ABCD, AB + CDE, AC + BDE, AD + BCE, AE + BCD, BC + ADE, BD + ACE, BE + ACD, CD + ABE, CE + ABD, DE + ABC.

Factor E (Helper number). Helpers are workers in the production line who are not attached to any machine; they do not operate any machine but perform tasks such as bundle handling, trimming, separating bundles, transporting bundles, matching part, and manual attaching of rope to the trousers. Just like the machine number, the effect of increasing and decreasing helper number in the production line was studied i.e. three helpers were added and three were removed from the production line.

Hypothetically, there are main factors and their interactions that might influence the throughput of the assembly line. Therefore, Resolution-V experimental design was used to test this hypothesis. This is because resolution-V design has greater ability to allow all main effects and two-way interactions to be fitted [38]. It was used to study the effect of the selected five input factors on the response (throughput). The selection of the design method was based on the hypothesis that three factors and higher order interactions are insignificant [38]. The resolution-V design was developed using Minitab software (version 18, Minitab Inc., USA) with the design specifications as shown in Table 1.

The resolution-V design confounds main factors effects with four-factors interactions and two-factors interactions with three-factors interactions as represented in the alias structure. This implies that the model for resolution-V design can contain all of the main effects and two-factor interactions. While the three-factors and higher order interaction are rare, and so they were safely ignored [38]. The experimental design in coded values is presented in Table 2.

**Table 2. Experimental design table.**

| Run | Block | A | B | C | D | E |
|-----|-------|---|---|---|---|---|
| 1 | 1 | + | - | - | + | + |
| 2 | 1 | - | - | + | + | + |
| 3 | 1 | - | + | - | + | + |
| 4 | 1 | + | - | + | + | - |
| 5 | 1 | + | - | - | - | - |
| 6 | 1 | + | + | + | + | + |
| 7 | 1 | - | - | + | - | - |
| 8 | 1 | - | + | + | - | + |
| 9 | 1 | + | - | + | - | + |
| 10 | 1 | + | + | + | - | - |
| 11 | 1 | + | + | - | - | + |
| 12 | 1 | - | - | - | - | + |
| 13 | 1 | - | + | + | + | - |
| 14 | 1 | - | + | - | - | - |
| 15 | 1 | - | - | - | + | - |
| 16 | 1 | + | + | - | + | - |

A, B, C, D and E denote factors; − and + are levels (low and high)

**Table 3. The design scenarios (training dataset).**

| Design scenario | Factors | | | | | Average throughput (pieces per day) |
|---|---|---|---|---|---|---|
| | A | B | C | D | E | |
| 1 | 40 | No policy | Random | Increase | Increase | 609 |
| 2 | 10 | No policy | Equal | Increase | Increase | 638 |
| 3 | 10 | Policy | Random | Increase | Increase | 583 |
| 4 | 40 | No policy | Equal | Increase | Reduce | 496 |
| 5 | 40 | No policy | Random | Reduce | Reduce | 465 |
| 6 | 40 | Policy | Equal | Increase | Increase | 607 |
| 7 | 10 | No policy | Equal | Reduce | Reduce | 467 |
| 8 | 10 | Policy | Equal | Reduce | Increase | 467 |
| 9 | 40 | No policy | Equal | Reduce | Increase | 467 |
| 10 | 40 | Policy | Equal | Reduce | Reduce | 467 |
| 11 | 40 | Policy | Random | Reduce | Increase | 429 |
| 12 | 10 | No policy | Random | Reduce | Increase | 467 |
| 13 | 10 | Policy | Equal | Increase | Reduce | 496 |
| 14 | 10 | Policy | Random | Reduce | Reduce | 439 |
| 15 | 10 | No policy | Random | Increase | Reduce | 496 |
| 16 | 40 | Policy | Random | Increase | Reduce | 496 |

A = Bundle size, B = Job release policy, C = Task assignment pattern, D = Machine number, E = Helper number.

**Generation of training datasets or design scenarios.** After designing the simulation experiment using Minitab software, 16 runs or design points were the outcomes from the design of the experiment. The number of runs represented different design scenarios for the simulation model of the garment assembly line. Thus, 16 design scenarios (training datasets) were generated. The simulation experiments were performed on each design scenario with the same run length (1 month of 8 hours working days), warm-up period (2 days) and replication number, n = 10.

To this end, the complex simulation model of the garment assembly line was used to perform 16 experimental runs. The base simulation model was altered depending on 16 design points generated from the design of experiment. Therefore, 16 design scenarios were created from the simulation experiments (https://doi.org/10.17632/T5W96KH5W7.1) [47]. Experimental runs were performed on each design scenario while observing the mean throughput (Table 3).

**Model training/fitting and validation.** Statistical-based approach was used to develop linear regression metamodel for analyzing effects of factors on throughput of the garment production line in order to answer the following questions: Which factors are important? How do the factors influence the simulation response (throughput)? What are the possible interaction effects between factors? The basis of this effect analysis is on the design matrix as defined by the design of experiment (resolution-V design). This was accomplished by statistical analysis of the 16 training datasets (design scenarios). Two-way Analysis of variance (ANOVA) was performed using Minitab statistical software (version 18, Minitab Inc, USA). The fitted metamodel was checked to see if the fidelity is adequate for the intended use. For this study, a simple significance check was used to validate the regression metamodel [21].

## Results and discussion

### Linear regression metamodel

The linear regression metamodel was analyzed using regression analysis in Minitab. Table 4 shows the factorial regression analysis of the response (throughput) versus factors: bundle size, job release policy, task assignment pattern, machine number and helper number for the 16 training datasets.

The results showed that the model has 15 DF; five (5) DF for Linear model and 10 DF for two-way interaction model. There was no DF for error for the designed model which implied that the observed response (throughput) value is equal to the model predicted throughput. This is because the average throughput from the simulation model has already been automatically fitted for the entire run length and replication numbers. Unlike physical experiments, computer experiments are deterministic hence there are no random errors for each replication [21]. In addition, the regression analysis of resolution-V design is always incomplete because the experiment is saturated, and all the available DF are consumed by the metamodel [38]. This resulted into no DF for residual error(s), and the adjusted mean square (Adj MS) of the error was not defined for the metamodel giving $R^2 = 1$. Further, the adjusted sums of squares (Adj SS) was also not defined for the error. Hence there was no residual plots for this metamodel design. These results showed a biased approximation of the simulation model. Nevertheless, it indicated that the metamodel is a good approximation of the simulation model since the mean square error (MSE) was equal to zero. The multiple linear regression metamodel is

**Table 4. ANOVA results for the design scenarios.**

| Source | DF | Contribution (%) | Adj SS | Adj MS |
|---|---|---|---|---|
| Model | 15 | 100.00 | 64529.8 | 4302.0 |
| Linear | 5 | 77.24 | 49845.3 | 9969.1 |
| Bundle size | 1 | 0.03 | 20.2 | 20.2 |
| Job release policy | 1 | 1.44 | 930.3 | 930.3 |
| Task assignment pattern | 1 | 1.44 | 930.2 | 930.2 |
| Machine number | 1 | 55.06 | 35532.2 | 35532.2 |
| Helper number | 1 | 19.27 | 12432.3 | 12432.3 |
| 2-Way Interactions | 10 | 22.76 | 14684.5 | 1468.4 |
| Bundle size*Job release policy | 1 | 0.20 | 132.2 | 132.2 |
| Bundle size*Task assignment pattern | 1 | 0.20 | 132.2 | 132.2 |
| Bundle size*Machine number | 1 | 0.00 | 2.3 | 2.3 |
| Bundle size*Helper number | 1 | 0.47 | 306.2 | 306.2 |
| Job release policy*Task assignment pattern | 1 | 0.33 | 210.2 | 210.2 |
| Job release policy*Machine number | 1 | 0.00 | 2.2 | 2.2 |
| Job release policy*Helper number | 1 | 0.47 | 306.2 | 306.2 |
| Task assignment pattern*Machine number | 1 | 0.02 | 12.2 | 12.2 |
| Task assignment pattern*Helper number | 1 | 0.37 | 240.2 | 240.2 |
| Machine number*Helper number | 1 | 20.67 | 13340.3 | 13340.3 |
| Error | 0 | * | * | * |
| Total | 15 | 100.00 | | |

DF- Degrees of Freedom, AdjSS- Adjusted sums of squares, AdjMS- Adjusted mean square

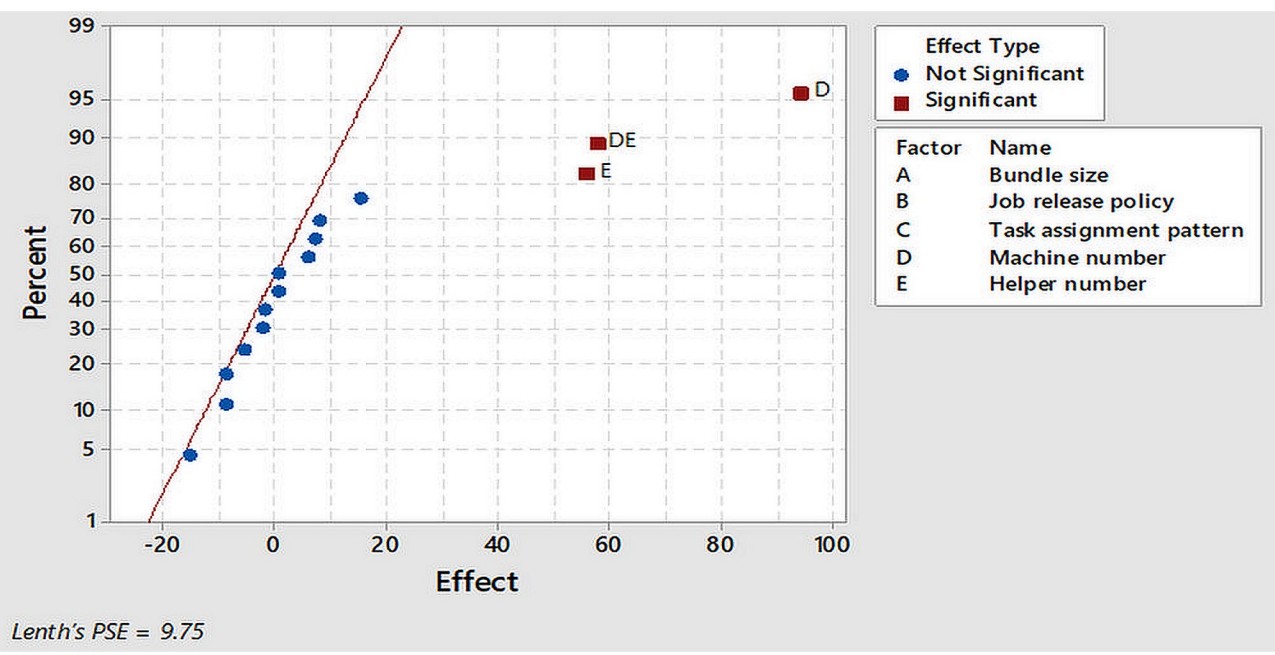

**Fig 8. Normal plot of the effects.**

presented in Eq 1. It is a first-order polynomial with 5 linear terms and two-way interactions.

$$Throughput = 507.5 - 0.075A - 12.42B + 12.42C + 46.5D + 35.17E + 0.1917AB \quad (1)$$
$$- 0.1917AC + 0.025AD - 0.2917AE + 3.625BC + 0.375BD - 4.375BE$$
$$- 0.875CD + 3.875CE + 28.88DE$$

A simple significance check was used to validate the linear regression metamodel. The presence of significant factors validated the metamodel. The plots in Fig 8 indicate that there are many terms (factors and their interactions) of near zero effect which were considered insignificant. This is because at 95% confidence interval ($\alpha = 0.05$) and Lenth's pseudo standard error (PSE) = 9.75, it is assumed that variation in the smallest effects are due to random errors. However, the outliers (E, DE and D) were considered significant because they have a large effect on the throughput. By applying Pareto analysis, insignificant terms were removed from the metamodel. Fig 9 illustrates that only the effect of the terms: two main factors (D and E) and one interaction (DE) exceeded the reference line at effect level (25.1) with Lenth's PSE = 9.75. These were retained in the regression metamodel while the terms having effects below the reference line were safely removed. In this respect, a new linear regression metamodel including two inputs (D and E) with two-way interaction (DE) was obtained (Eq 2).

$$Throughput = 507.5 + 46.5D + 35.17E + 28.88DE \quad (2)$$

The new linear regression metamodel presented could be adopted for predictions without going through complex simulation experiments. However, it did not work in this study. This is because of the different resource types and the large number of workstations involved in the trouser assembly line system. It would be impractical merely feeding the machine numbers or helper numbers into Eq 2 to determine the average throughput without undergoing the complex simulation experiments. Using metamodel for prediction is suitable for single machine

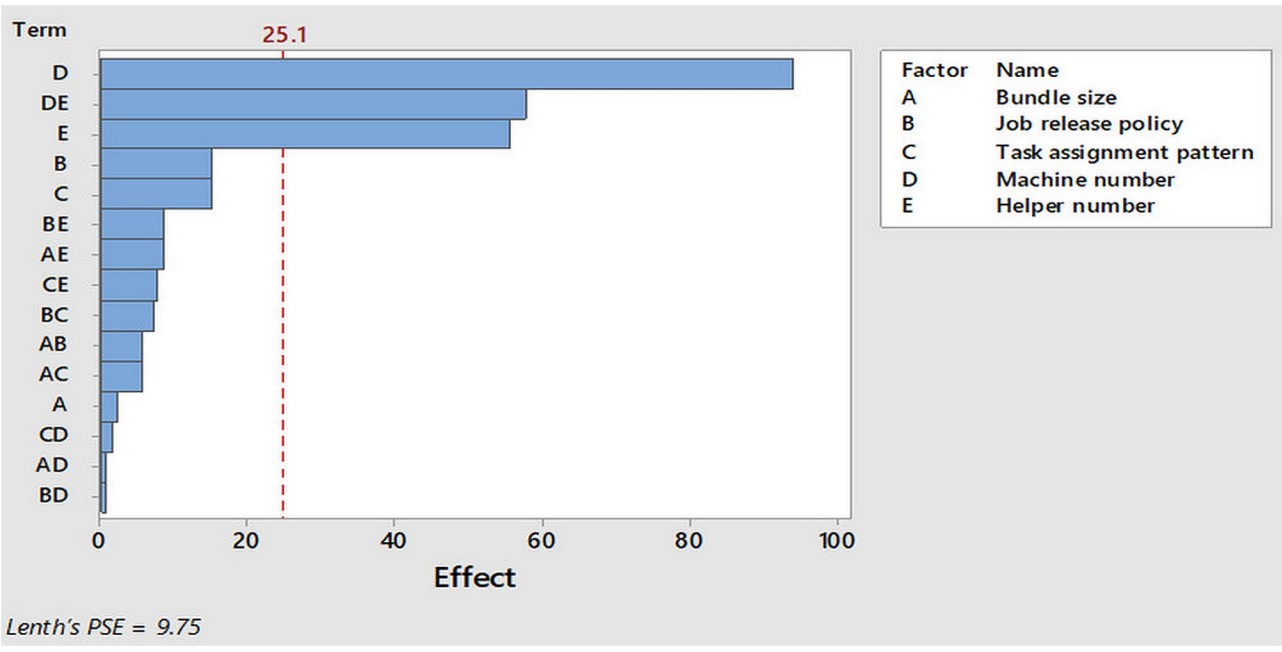

**Fig 9. Pareto chart of the effects.**

shop but becomes impractical for application in assembly line system (a series of different and identical machines) [39]. For this reason, the best parameters' setting (Table 5) with the highest throughput was selected from the training dataset. A bundle size of 25 was adopted instead of 10, because varying bundle sizes had insignificant effect on the mean throughput (see explanation under **the main factors effect**). Moreover, using bundle size of 10 produced high idle times for the preparatory sections. Hence the normal bundle size of 25 was the most suitable for the trouser assembly line design. The best parameter's setting achieved an average throughput of 638 pieces per day, resulting in a 28.63% increase in the production throughput of the existing design (which has an average throughput of 490 pieces per day). This is in congruence with Atan et al. [51] who reported that increasing resources in the bottleneck workstations increases average throughput. This is because it reduces both cycle time and WIP. In spite of the fact that the metamodel was not used for model prediction, the present study found it most suitable for inference [30], and for initial solution (local optimal) for optimization process [52]. The inference was drawn from the linear regression metamodel to analyze the factors' effects on throughput. This is very useful for line production planning because being insightful on the effect of factors can enhance decision making on which factors to consider so as to improve production throughput and efficiency.

**Table 5. The best parameters' setting of the metamodel.**

| S/N | Decision variables | Setting |
|---|---|---|
| 1 | Bundle size | 25 |
| 2 | Job release policy | No policy |
| 3 | Task assignment pattern | Equal |
| 4 | Machine number | Increase (1 iron press and 3 single needle lockstitch) |
| 5 | Helper number | Increase (3 helpers) |

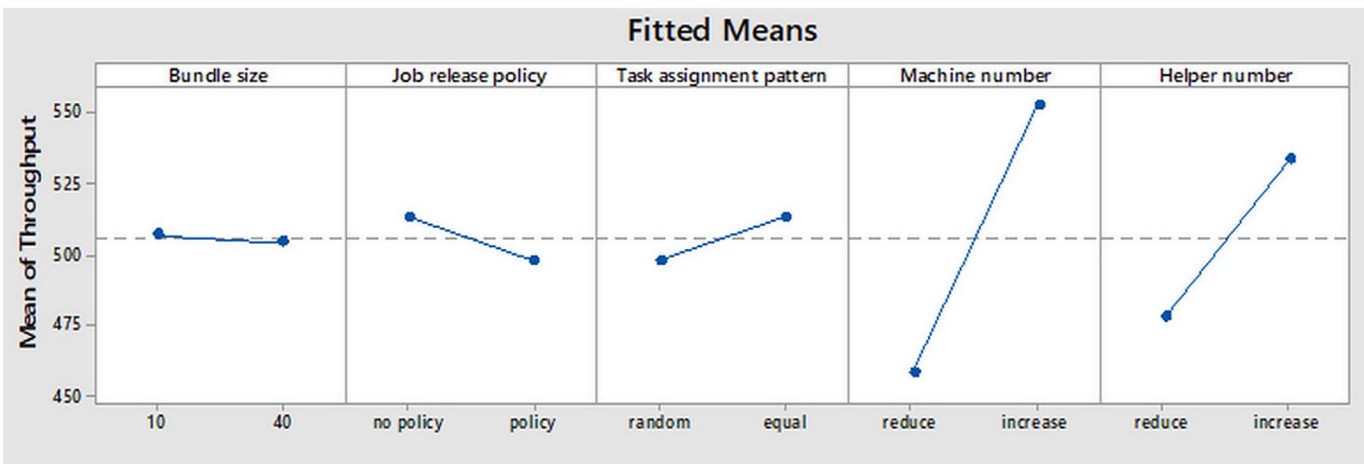

**Fig 10. Main factors effect plot for throughput.**

## The main factors effect

The main factors effect of the five factors (Bundle size, job release policy, task assignment pattern, machine number and helper numbers) on the production throughput are shown in Fig 10 and interpreted as follows.

**Bundle size effect plot.** With all factors kept constant, the mean throughput decreased by a very small value when the bundle size was changed from 10 to 40. Thus, if the same quantity of input materials were kept constant for all levels of bundle sizes, a very small decrease would be observed in the mean throughput when the bundle size of 40 is used. This is explained by the longer time it takes for each preparation section to complete tasks on bundles while keeping the main body assembly idle. This implies a longer warm up time for the production line resulting into low throughput.

**Job release policy effect plot.** The plot connotes that if all other factors were kept constant, changes in the level of job release policy would have a greater change in the mean throughput when compared to that of the bundle size. A decrease in the mean throughput was observed when the job release policy was changed from no policy to policy level. This is because at no policy level, the quantity of input materials is kept constant in the production line and every preparation section is capable of preparing enough parts for the main body assembly section. As for the case of policy system which was based on the WIP threshold of the bottleneck workstation, there is a lot variability in the throughput of the different sections as they have to wait for the input materials and thus, affecting the productivity of the main body assembly. This therefore reduces the overall throughput of the production line. For instance, big loop preparation has to be done at a faster rate than other sub-assembly processes because seven loops are required to be assembled on one trouser. For this reason, any delays in the preparation process could cause starvation of the main body assembly as well as the extreme workstations resulting into low throughput. In previous studies, job release policy based on WIP threshold of the bottleneck workstation was observed to increase throughput [53, 54]. In contrast, the present study achieved lower throughput. A plausible explanation is that previous studies considered the assembly line problem which does involve parts preparation processes. Consequently, keeping WIP of one workstation does not starve the extreme workstations, thus increasing the throughput. It should be emphasized that job release policy based on WIP threshold of the bottleneck workstation does not work well on the assembly line

problem that requires part preparation process as it leads to starvation of the main body assembly resulting into low throughput.

**Task assignment pattern effect plot.** There was a small increment in the average throughput when random task assignment pattern was changed to equal task assignment. With the random task assignment, there is unequal workload for operators performing similar tasks in the workstation. Thus, the workstation cycle and idle times are increased, resulting into low throughput. On the other hand, equal task assignment maintains the same workload among operators, reducing the cycle and idle times which ultimately increases the overall throughput. The present study is in complete agreement with the report of Kandemir & Handley [55] who reiterated that equal task assignment had higher production throughput and efficiency due to equal workload of the operators and minimization of workstation idle time.

**Machine number effect plot.** The effect of machine numbers on the throughput was found to be statistically significant at $\alpha = 0.05$. This means that, the throughput increases when machine number is increased in the workstation and vice versa. This is because increasing machine number in the bottleneck workstation reduces cycle and parts waiting times as well as the WIP.

**Helper number effect plot.** Similarly, helper number had a significant effect on the mean throughput though its effect was smaller when compared to machine number. The work of helpers in the production line normally influences the feeding of parts to the extreme workstations. When the number of helpers is increased, the extreme workstation is never starved of materials due to reduction of helper's workstation cycle time and WIP. Subsequently, a higher throughput is realized. Contrastingly, decreasing the number of helper results in an increase in their workstation cycle time, leading to starvation of the extreme workstations thus a lower throughput.

## The interaction effects

The interaction effect of the factors on the production throughput is illustrated in Fig 11. Each plot represents the interaction between two factors. When the red and blue lines of the factor levels are with considerably different slopes, it indicates that there is an interaction between the two factors. In this respect, the interaction effects of the factors were interpreted as follows.

**Bundle size and job release policy effect plot.** This interaction plot indicated that there is very little interaction between bundle size and job release policy as the no policy and policy lines took slightly different slopes. With the bundle size of 10, the average throughput decreased by a large value when the job release policy was changed from no policy to policy level. While with the bundle size of 40, the throughput decreased by a very small value and almost did not change at all when the job release policy was changed from no policy to policy level.

**Bundle size and task assignment pattern effect plot.** There was also an insignificant effect of the interaction between bundle size and task assignment. Nonetheless, there was little interaction effect as the slopes of random and equal lines are not parallel. This implies that with the bundle size of 10, the mean throughput increased by a large value when task assignment pattern changed from random to equal level. With the bundle size at 40, the mean throughput increased by a very small value when the task assignment pattern changed from random to equal level.

**Bundle size and machine number effect plot.** There was actually no interaction between bundle size and the number of machines since there was no significant difference in the slopes of the reduce and increase levels considered. This points out that there could not be any differences even if the alpha value were increased.

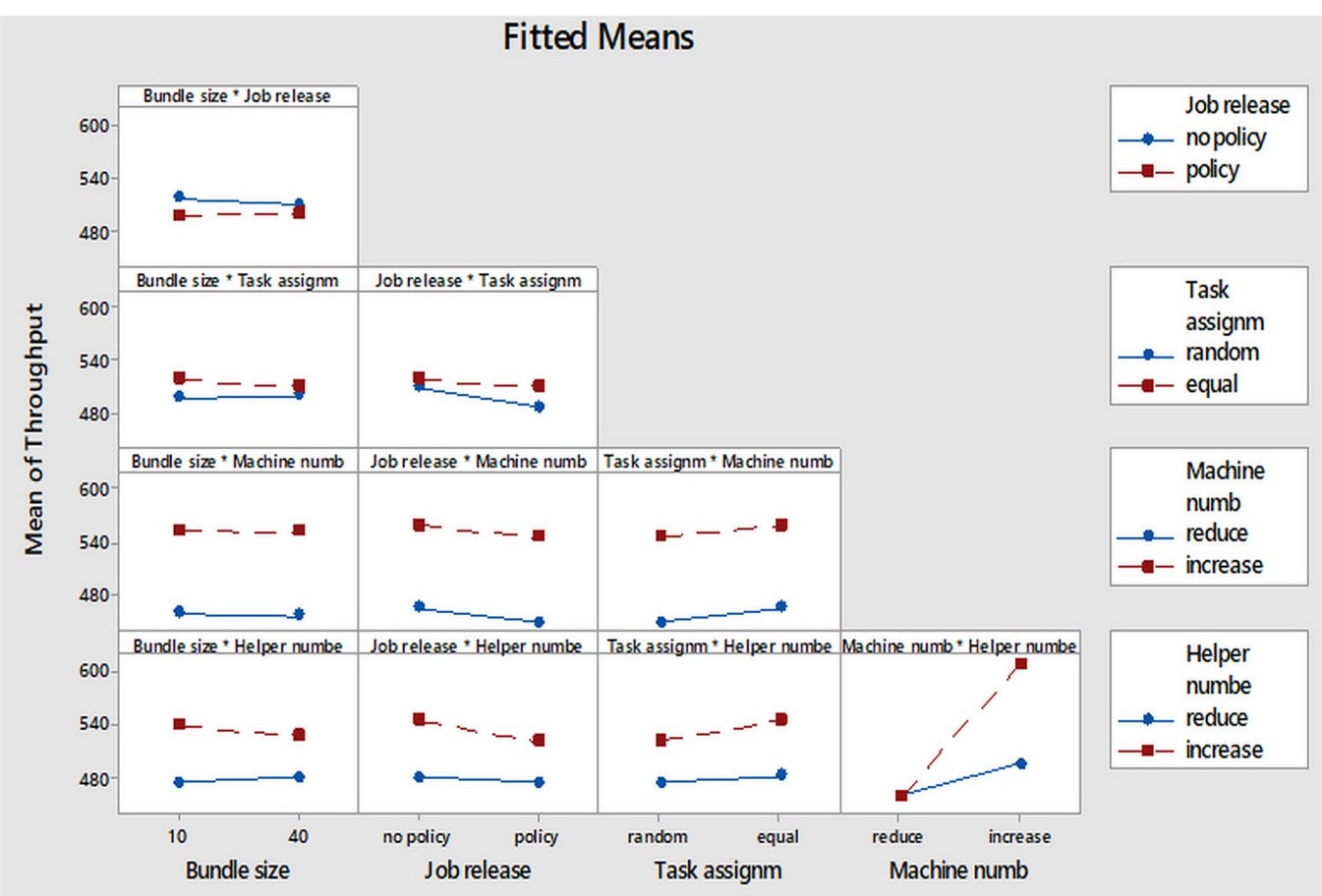

**Fig 11. Interaction plot for throughput.**

**Bundle size and helper number effect plot.** The slope of the reduce and increase lines of the two levels of the helper numbers differed slightly. The plot indicates that if the bundle size is at 10, the mean throughput increases by a larger value when the helper number is changed from reduce to increase level (helper number is increased). The reverse would be true if the bundle size is at 40.

**Job release policy and task assignment pattern effect plot.** The slope of random and equal lines of the two levels of task assignment pattern took slightly different directions. Consequently, significant interactions can exist when the alpha value is increased. The plot connotes that, if the job release policy is at no policy, there is almost no change or a very small increase in the throughput when the task assignment pattern changes from random to equal level. But if the job release policy is set at policy level, the throughput increases by a bigger value when the task assignment pattern changes from random to equal level.

**Job release policy and machine number effect plot.** There is likely to be no interaction between job release policy and machine number at all even if the alpha value is further increased because the slope of the reduce and increase levels of machine number are parallel.

**Job release policy and helper number effect plot.** There is an insignificant interaction between job release policy and helper number at $\alpha = 0.05$. Notwithstanding, the reduce and increase levels of helper number took slightly different slopes. Thus, their interaction could be significant when the alpha value is further increased at some point.

**Task assignment pattern and machine number effect plot.** There is no interaction between task assignment and machine number as the slope of the reduce and increase levels of machine number took the same direction.

**Task assignment pattern and helper number effect plot.** It can be observed from the plot that the slope of the reduce and increase lines of helper number are parallel. The plot therefore means that, if the task assignment pattern is at random level, the average throughput insignificantly increases when the helper number is increased. However, if the task assignment pattern is at equal level, the average throughput increases by a larger value when the helper number is increased.

**Machine number and helper number effect plot.** There was a statistically significant interaction between machine number and helper number at $\alpha = 0.05$. It is observed that the slope of the reduce and increase lines of helper number differed significantly. The plot means that if the machine number is at reduce level, there is no change in the mean throughput when the helper number is increased. The reverse is true when the machine number is at increase level. It can be noted that increasing helper number when the machine number is at reduce level does not change the average throughput because helpers perform simple tasks in the production line and their tasks depends on the workstations with machines. Reducing helper number contributes to high WIP and idle time in the workstations for the helpers but the throughput remains constant because only the cycle time of the helpers is changed. However, increasing the helper number when the machine number is at increase level increases throughput because the cycle and idle times as well as the WIP of both the machine and the helper workstations are reduced.

## Conclusion

The present study demonstrated a garment assembly line design using simulation metamodeling with 28.63% increase of the throughput achieved for the best setting of the metamodel. However, the developed metamodel was a biased approximation of trouser assembly line simulation model with $R^2 = 1$ and MSE = 0. This metamodel is not suitable for prediction but rather for inference because of the complex nature of the assembly line which is composed of many workstations with both identical and different machines. The metamodel was used to give an insight of the relationship between factors (bundle size, job release policy, task assignment pattern, machine number and helper number) and the throughput, identifying the most influential factors and quantifying their impact on the throughput and detecting important interactions. The job release policy based on the WIP threshold of the bottleneck workstation is not suitable for improving the throughput of assembly line with parts preparation process (sub-assembly process). In order to overcome the biasness of the metamodel, a further study should use a space-filling experimental design such as Latin hypercube design or orthogonal array. Further, profound metamodeling technique involving machine learning approach for designing garment assembly line should be investigated.

## Supporting information

**S1 Fig. Arena animation of trouser assembly line.**
(TIF)

**S2 Fig. Arena crystal report for average throughput.**
(TIF)

**S1 Table. Fitted processing times probability distribution for bundle size-10.** [a]OPN- Operation Number, [d]S/NL- Single needle lockstitch, [b]BH- Button hole machine, [h]F/A- Feed of arm,

[c]O/L- overlock machine, [k]LM- loop stitching machine, [i]D/NL- Double needle lockstitch, [j]BT-Bartack machine, [g]TM- turning machine, [f]AWM- automatic wallet machine, [e]F/L- Flatlock machine. This legend applies for both S2 and S3 Tables.
(DOCX)

**S2 Table. Fitted processing times probability distribution for bundle size—25.**
(DOCX)

**S3 Table. Fitted processing times probability distribution for bundle size—40.**
(DOCX)

**S4 Table. The throughput sample data from real garment assembly line system.**
(DOCX)

**S5 Table. Daily input material for real garment assembly line system.**
(DOCX)

**S1 File. The video of running Arena simulation model of trouser assembly line.**
(MP4)

## Acknowledgments

The authors acknowledge the Rockwell Automation for providing Arena Software Academic Research License used in this study. The authors are also grateful to the management of Southern Range Nyanza Limited (NYTIL) for allowing the research to be conducted in their facility. Sincere thanks are due to Timothy Omara (Department of Chemistry and Biochemistry, Moi University) for the technical advices and English proofreading of this work.

## Author Contributions

**Conceptualization:** Ocident Bongomin, Eric Oyondi Nganyi.

**Data curation:** Ocident Bongomin, Eric Oyondi Nganyi.

**Formal analysis:** Ocident Bongomin, Josphat Igadwa Mwasiagi.

**Funding acquisition:** Josphat Igadwa Mwasiagi, Eric Oyondi Nganyi.

**Investigation:** Ocident Bongomin, Eric Oyondi Nganyi, Ildephonse Nibikora.

**Methodology:** Ocident Bongomin.

**Project administration:** Ocident Bongomin, Josphat Igadwa Mwasiagi.

**Resources:** Ocident Bongomin, Josphat Igadwa Mwasiagi.

**Software:** Ocident Bongomin.

**Supervision:** Josphat Igadwa Mwasiagi, Eric Oyondi Nganyi, Ildephonse Nibikora.

**Validation:** Ocident Bongomin, Ildephonse Nibikora.

**Visualization:** Ocident Bongomin, Josphat Igadwa Mwasiagi, Ildephonse Nibikora.

**Writing – original draft:** Ocident Bongomin.

**Writing – review & editing:** Ocident Bongomin, Josphat Igadwa Mwasiagi, Eric Oyondi Nganyi.

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
