## [Decision Letter · Decision Letter 0]

1 Jul 2020

PONE-D-20-13481

Simulation metamodeling approach to complex design of garment assembly lines

PLOS ONE

Dear Dr. Bongomin,

Thank you for submitting your manuscript to PLOS ONE. After careful consideration, we feel that it has merit but does not fully meet PLOS ONE’s publication criteria as it currently stands. Therefore, we invite you to submit a revised version of the manuscript that addresses the points raised during the review process.

We look forward to receiving your revised manuscript.

Kind regards,

Ziqiang Zeng, Ph.D.

Academic Editor

PLOS ONE

Journal Requirements:

3. We note that in the Data Availability statement you have provided a link to the dataset used in this study. We ask that you additionally provide this link within the Methods section for the convenience of the reader.

Additional Editor Comments (if provided):

Based on the reviewers' comments, I suggest the authors to do a major revision for this paper and submit a detailed response, so we can further consider if it is suitable to accept this paper.

Reviewers' comments:

Reviewer's Responses to Questions

**Comments to the Author**

1. Is the manuscript technically sound, and do the data support the conclusions?

Reviewer #1: Yes

Reviewer #2: Partly

2. Has the statistical analysis been performed appropriately and rigorously? 

Reviewer #1: Yes

Reviewer #2: Yes

3. Have the authors made all data underlying the findings in their manuscript fully available?

Reviewer #1: Yes

Reviewer #2: No

4. Is the manuscript presented in an intelligible fashion and written in standard English?

Reviewer #1: Yes

Reviewer #2: Yes

5. Review Comments to the Author

Reviewer #1: (1) The paper is well written and includes interesting findings for the engineers. I do not have any specific comment, however, I recommend improving the figures quality, and writing level. Also please clearly specify the contribution at the end of introduction section. Try to condense the theoretical parts and use the flowcharts as much as possible for illustrative purposes.

(2) Literature review must include application of Latin hypercube design and orthogonal array for design of experiment. Further, if profound metamodeling technique involving machine learning approach is used in this domain must be cited in introduction of paper.

Reviewer #2: I believe this paper presents an interesting piece of work. I am not particularly very familiar with the garment manufacturing and research literature, as I am with the simulation metamodeling framework in general. The authors mention that simulation is a commonly applied tool within the garment and assembly line industries. However, the same does not hold for simulation metamodels. For this reason, i.e., for using a technique that is not (yet) often applied within a particular field, I think that this work already has some merit.

On the other hand, there is much room for improvement.

First of all, the pictures/figures used. I found two problems: their bad quality or low resolution. For instance, I can barely read the contents of Figure 2. Secondly: I do not find it very comfortable for the figures to be at the end of the paper. The constant scroll up/down between the figures and the text where they are referenced is very distracting and demotivating. The authors should try to solve these two problems.

The paper is generally well written, but the writing style and some typos should be reviewed here and there. Some examples (with correction/suggestion): "However, most of THE simulation studies have considered ill-defined experimental designS..."; "Simulation has been majorly applied for THE analysis of complex systems to give extensive insightS..." (?)

Here "Metamodels, also well-known as meta-models, surrogate models, or emulators are used to approximate the input-output behavior of simulation models.", I would remove the reference to "meta-models" since metamodels = meta-models (it is just a different way of writing the same thing, although I think "metamodel" is more common). I would add "response surfaces" too.

In line 62, the authors mention that metamodeling aims to reduce the computation costs of simulation models during the optimization process. This is only partially true. It is true if the metamodelling strategy is being employed within the simulation-optimization approaches, but it is not valid if there is no optimization process involved. In general, simulation metamodels aim to bypass the simulation model itself yes!

This paper "Jack PC Kleijnen and Wim CM Van Beers. Application-driven sequential designs for simulation experiments: Kriging metamodelling. Journal of the Operational Research Society, 55(8):876–883, 2004.", elegantly identifies the four primary objectives of simulation metamodels. Speaking of other papers, I think that the literature review around simulation models could be expanded a bit, as it is the central theme of this work.

I could understand that the Fractional Factorial design was used, but it not very clear to me what type of regression model was used as metamodel. Was it the Multiple Linear Regression model? I could infer this from Equations 1 and 2.

These are my comments for now.

6. PLOS authors have the option to publish the peer review history of their article (what does this mean?). If published, this will include your full peer review and any attached files.

Reviewer #1: **Yes: **Ramin Ghiasi

Reviewer #2: No

---

## [Author Response · Author response to Decision Letter 0]

9 Jul 2020

JOURNAL REQUIREMENTS:

Thank you for this recommendation. We have revised the manuscript as per the suggested journal style requirements.

We suggest you thoroughly copyedit your manuscript for language usage, spelling, and grammar. If you do not know anyone who can help you do this, you may wish to consider employing a professional scientific editing service. Whilst you may use any professional scientific editing service of your choice, PLOS has partnered with both American Journal Experts (AJE) and Editage to provide discounted services to PLOS authors. Both organizations have experience helping authors meet PLOS guidelines and can provide language editing, translation, manuscript formatting, and figure formatting to ensure your manuscript meets our submission guidelines. To take advantage of our partnership with AJE, visit the AJE website (http://learn.aje.com/plos/) for a 15% discount off AJE services. To take advantage of our partnership with Editage, visit the Editage website (www.editage.com) and enter referral code PLOSEDIT for a 15% discount off Editage services. If the PLOS editorial team finds any language issues in text that either AJE or Editage has edited, the service provider will re-edit the text for free.

The manuscript has been checked for grammatical fixes and English language by a proficient colleague.

The manuscript was proofread by Timothy Omara, an experienced researcher with the Department of Chemistry and Biochemistry, Moi University, Kenya. We have added this to the acknowledgement section of the revised manuscript. 

We have uploaded a copy of the manuscript with highlighted changes as a supporting information file.

This has been uploaded as suggested. 

2. We note that in the Data Availability statement you have provided a link to the dataset used in this study. We ask that you additionally provide this link within the Methods section for the convenience of the reader.

Thank you for this suggestion. We have included the link to the dataset in the methodology section (page 9, Line 191 and page 14, Line 277).

ADDITIONAL EDITOR COMMENTS (IF PROVIDED):

Based on the reviewers' comments, I suggest the authors to do a major revision for this paper and submit a detailed response, so we can further consider if it is suitable to accept this paper.

We have done extensive revision of the manuscript closely following the reviewers’ comments. Thank you.

REVIEWERS' COMMENTS:

Reviewer's Responses to Questions

1. Is the manuscript technically sound, and do the data support the conclusions?

Reviewer #1: Yes

Reviewer #2: Partly

2. Has the statistical analysis been performed appropriately and rigorously?

Reviewer #1: Yes

Reviewer #2: Yes

3. Have the authors made all data underlying the findings in their manuscript fully available?

Reviewer #1: Yes

Reviewer #2: No

4. Is the manuscript presented in an intelligible fashion and written in standard English?

Reviewer #1: Yes

Reviewer #2: Yes

5. Review Comments to the Author

Reviewer #1: 

(1) The paper is well written and includes interesting findings for the engineers. I do not have any specific comment, however, I recommend improving the figures quality, and writing level. Also please clearly specify the contribution at the end of introduction section. Try to condense the theoretical parts and use the flowcharts as much as possible for illustrative purposes.

Thank you for the insightful comments. We have improved the resolution of the figures. The manuscript have been proofread by a proficient colleague. The contribution of our study has also been added to the introduction section as advised (page 6, Lines 124-126)

We have included a flowchart (Fig 1; page 7) in the methodology section to further describe the approach used in the study. 

(2) Literature review must include application of Latin hypercube design and orthogonal array for design of experiment. Further, if profound metamodeling technique involving machine learning approach is used in this domain must be cited in introduction of paper.

This is quite correct. We have included the application of Latin hypercube design and orthogonal array on page 5, Lines107-109 (introduction section). We have also cited the use of machine learning technique (e.g. Artificial neural network) for metamodeling on page 5, Lines 94-96.

Reviewer #2: 

I believe this paper presents an interesting piece of work. I am not particularly very familiar with the garment manufacturing and research literature, as I am with the simulation metamodeling framework in general. The authors mention that simulation is a commonly applied tool within the garment and assembly line industries. However, the same does not hold for simulation metamodels. For this reason, i.e., for using a technique that is not (yet) often applied within a particular field, I think that this work already has some merit.

Thank you for this commendation. 

On the other hand, there is much room for improvement.

First of all, the pictures/figures used. I found two problems: their bad quality or low resolution. For instance, I can barely read the contents of Figure 2. Secondly: I do not find it very comfortable for the figures to be at the end of the paper. The constant scroll up/down between the figures and the text where they are referenced is very distracting and demotivating. The authors should try to solve these two problems.

Firstly, we apologize for this inconvenience. Higher resolution figures have been supplied. For the positioning of the figures, these were done in accordance with PLoS Ones’ manuscript preparation guidelines. For this reason, we could not reposition these figures. Thank you. 

The paper is generally well written, but the writing style and some typos should be reviewed here and there. Some examples (with correction/suggestion): "However, most of THE simulation studies have considered ill-defined experimental designS..."; "Simulation has been majorly applied for THE analysis of complex systems to give extensive insightS..." (?)

This is quite correct. The manuscript has been extensively revised to eliminate typos, grammatical and spelling errors.

Here "Metamodels, also well-known as meta-models, surrogate models, or emulators are used to approximate the input-output behavior of simulation models.", I would remove the reference to "meta-models" since metamodels = meta-models (it is just a different way of writing the same thing, although I think "metamodel" is more common). I would add "response surfaces" too.

Thank you for this expert suggestion. We removed meta-models and added response surfaces as suggested.

In line 62, the authors mention that metamodeling aims to reduce the computation costs of simulation models during the optimization process. This is only partially true. It is true if the metamodeling strategy is being employed within the simulation-optimization approaches, but it is not valid if there is no optimization process involved. In general, simulation metamodels aim to bypass the simulation model itself yes!

Very well, we have removed this statement and included the general advantages of metamodeling in optimization (page 4 line 86 and page 5 lines 87-88)

This paper "Jack PC Kleijnen and Wim CM Van Beers. Application-driven sequential designs for simulation experiments: Kriging metamodeling. Journal of the Operational Research Society, 55(8):876–883, 2004.", elegantly identifies the four primary objectives of simulation metamodels. Speaking of other papers, I think that the literature review around simulation models could be expanded a bit, as it is the central theme of this work.

We have reviewed more papers including the suggested paper and others on simulation models. We have expanded this part of the manuscript (please check page 3, Lines 53-64 and page 5, Lines 91-96). We hope this is agreeable. 

I could understand that the Fractional Factorial design was used, but it is not very clear to me what type of regression model was used as metamodel. Was it the Multiple Linear Regression model? I could infer this from Equations 1 and 2.

You are very right. It is a multiple linear regression model with linear terms and two-way interactions that we used. Thank you

These are my comments for now.

6. PLOS authors have the option to publish the peer review history of their article (what does this mean?). If published, this will include your full peer review and any attached files.

Do you want your identity to be public for this peer review? For information about this choice, including consent withdrawal, please see our Privacy Policy.

Reviewer #1: Yes: Ramin Ghiasi

Reviewer #2: No

---

## [Decision Letter · Decision Letter 1]

26 Aug 2020

PONE-D-20-13481R1

Simulation metamodeling approach to complex design of garment assembly lines

PLOS ONE

Dear Dr. Bongomin,

Thank you for submitting your manuscript to PLOS ONE. After careful consideration, we feel that it has merit but does not fully meet PLOS ONE’s publication criteria as it currently stands. Therefore, we invite you to submit a revised version of the manuscript that addresses the points raised during the review process.

We look forward to receiving your revised manuscript.

Kind regards,

Ziqiang Zeng, Ph.D.

Academic Editor

PLOS ONE

Additional Editor Comments (if provided):

Based on the review comments, I would like to accept this paper after you make a minor revision according to the reviewer's comments.

Reviewers' comments:

Reviewer's Responses to Questions

**Comments to the Author**

1. If the authors have adequately addressed your comments raised in a previous round of review and you feel that this manuscript is now acceptable for publication, you may indicate that here to bypass the “Comments to the Author” section, enter your conflict of interest statement in the “Confidential to Editor” section, and submit your "Accept" recommendation.

Reviewer #1: All comments have been addressed

Reviewer #2: All comments have been addressed

2. Is the manuscript technically sound, and do the data support the conclusions?

Reviewer #1: Yes

Reviewer #2: Partly

3. Has the statistical analysis been performed appropriately and rigorously? 

Reviewer #1: Yes

Reviewer #2: Yes

4. Have the authors made all data underlying the findings in their manuscript fully available?

Reviewer #1: Yes

Reviewer #2: No

5. Is the manuscript presented in an intelligible fashion and written in standard English?

Reviewer #1: Yes

Reviewer #2: Yes

6. Review Comments to the Author

Reviewer #1: (No Response)

Reviewer #2: My comments have been addressed.

I recommended the paper "Jack PC Kleijnen and Wim CM Van Beers. Application-driven sequential designs for simulation experiments: Kriging metamodeling. Journal of the Operational Research Society, 55(8):876–883, 2004." stating that it explains the four main objectives of simulation metamodels, but I made a mistake. It can still be used for the literature review, of course.

Although from the same first author, the paper that I had in mind and I was recommending it was the following: "Jack PC Kleijnen and Robert G Sargent. A methodology for fitting and validating metamodels in simulation. European Journal of Operational Research, 120(1):14–29, 2000." As the title itself indicates, this paper provides a more general and comprehensive overview of the simulation metamodelling process. Apologies for this mix-up.

7. PLOS authors have the option to publish the peer review history of their article (what does this mean?). If published, this will include your full peer review and any attached files.

Reviewer #1: No

Reviewer #2: No

---

## [Author Response · Author response to Decision Letter 1]

27 Aug 2020

ADDITIONAL EDITOR COMMENTS (IF PROVIDED):

Based on the review comments, I would like to accept this paper after you make a minor revision according to the reviewer's comments.

We have revised the manuscript closely to the reviewer’s comments. Thank you

REVIEWERS' COMMENTS:

Reviewer's Responses to Questions

Comments to the Author

1. If the authors have adequately addressed your comments raised in a previous round of review and you feel that this manuscript is now acceptable for publication, you may indicate that here to bypass the “Comments to the Author” section, enter your conflict of interest statement in the “Confidential to Editor” section, and submit your "Accept" recommendation.

Reviewer #1: All comments have been addressed

Reviewer #2: All comments have been addressed

2. Is the manuscript technically sound, and do the data support the conclusions?

Reviewer #1: Yes

Reviewer #2: Partly

3. Has the statistical analysis been performed appropriately and rigorously?

Reviewer #1: Yes

Reviewer #2: Yes

4. Have the authors made all data underlying the findings in their manuscript fully available?

Reviewer #1: Yes

Reviewer #2: No

5. Is the manuscript presented in an intelligible fashion and written in standard English?

Reviewer #1: Yes

Reviewer #2: Yes

6. Review Comments to the Author

Reviewer #1: (No Response)

Reviewer #2: My comments have been addressed.

I recommended the paper "Jack PC Kleijnen and Wim CM Van Beers. Application-driven sequential designs for simulation experiments: Kriging metamodeling. Journal of the Operational Research Society, 55(8):876–883, 2004." stating that it explains the four main objectives of simulation metamodels, but I made a mistake. It can still be used for the literature review, of course.

Although from the same first author, the paper that I had in mind and I was recommending it was the following: "Jack PC Kleijnen and Robert G Sargent. A methodology for fitting and validating metamodels in simulation. European Journal of Operational Research, 120(1):14–29, 2000." As the title itself indicates, this paper provides a more general and comprehensive overview of the simulation metamodelling process. Apologies for this mix-up.

We have included the suggested reference in the revised manuscript. Please check page 5, line 98-100. Thank you

7. PLOS authors have the option to publish the peer review history of their article (what does this mean?). If published, this will include your full peer review and any attached files.

Do you want your identity to be public for this peer review? For information about this choice, including consent withdrawal, please see our Privacy Policy.

Reviewer #1: No

Reviewer #2: No

---

## [Editor Report · Decision Letter 2]

7 Sep 2020

Simulation metamodeling approach to complex design of garment assembly lines

PONE-D-20-13481R2

Dear Dr. Bongomin,

We’re pleased to inform you that your manuscript has been judged scientifically suitable for publication and will be formally accepted for publication once it meets all outstanding technical requirements.

Kind regards,

Ziqiang Zeng, Ph.D.

Academic Editor

PLOS ONE

Additional Editor Comments (optional):

This paper is well revised. It can be accepted.
---

## [Editor Report · Acceptance letter]

10 Sep 2020

PONE-D-20-13481R2 

Simulation metamodeling approach to complex design of garment assembly lines 

Dear Dr. Bongomin:

I'm pleased to inform you that your manuscript has been deemed suitable for publication in PLOS ONE. Congratulations! Your manuscript is now with our production department. 

Kind regards, 

on behalf of

Dr. Ziqiang Zeng 

Academic Editor

PLOS ONE